# Maternal Supplementation of Food Ingredient (Prebiotic) or Food Contaminant (Mycotoxin) Influences Mucosal Immune System in Piglets

**DOI:** 10.3390/nu12072115

**Published:** 2020-07-17

**Authors:** Stéphanie Ferret-Bernard, Laurence Le Normand, Véronique Romé, Cindy Le Bourgot, Julie Seeboth, Gérard Savary, Fabrice Laurent, Isabelle Le Huërou-Luron, Laurence Guzylack-Piriou

**Affiliations:** 1INRAE, INSERM, University of Rennes, NuMeCan, Nutrition Metabolisms and Cancer, F-35000 Rennes, France; laurence.le-normand@inrae.fr (L.L.N.); veronique.rome@inrae.fr (V.R.); cindy.lebourgot@tereos.com (C.L.B.); gerard.savary@inrae.fr (G.S.); isabelle.luron@inrae.fr (I.L.H.-L.); 2Tereos, R&D Department, F-77230 Moussy-le-Vieux, France; 3UMR TOXALIM, University of Toulouse, INRA, ENVT, INP-Purpan, UPS, F-31027 Toulouse, France; julie.seeboth@inrae.fr; 4UMR ISP, INRAE, University of François Rabelais de Tours, F-37380 Nouzilly, France; fabrice.laurent@inrae.fr; 5IALTA, IHAP, University of Toulouse, ENVT, INRAE, F-31076 Toulouse, France; laurence.guzylack@inrae.fr

**Keywords:** pig, gut immunity, intestinal antigen-presenting cell, short-chain fructooligosaccharides, deoxynivalenol

## Abstract

The early life period is crucial for the maturation of the intestinal barrier, its immune system, and a life-long beneficial host–microbiota interaction. The study aims to assess the impact of a beneficial dietary (short-chain fructooligosaccharides, scFOS) supplementation vs. a detrimental dietary environment (such as mycotoxin deoxynivalenol, DON) on offspring intestinal immune system developmental profiles. Sows were given scFOS-supplemented or DON-contaminated diets during the last 4 weeks of gestation, whereas force-feeding piglets with DON was performed during the first week of offspring life. Intestinal antigen-presenting cell (APC) subset frequency was analyzed by flow cytometry in the Peyer’s patches and in *lamina propria* and the responsiveness of intestinal explants to toll-like receptor (TLR) ligands was performed using ELISA and qRT-PCR from post-natal day (PND) 10 until PND90. Perinatal exposure with scFOS did not affect the ontogenesis of APC. While it early induced inflammatory responses in piglets, scFOS further promoted the T regulatory response after TLR activation. Sow and piglet DON contamination decreased CD16+ MHCII+ APC at PND10 in *lamina propria* associated with IFNγ inflammation and impairment of Treg response. Our study demonstrated that maternal prebiotic supplementation and mycotoxin contamination can modulate the mucosal immune system responsiveness of offspring through different pathways.

## 1. Introduction

The early life period is crucial for the maturation of the intestinal barrier, its immune system, and a life-long beneficial host–microbiota interaction. From birth, the mucosal immune system is exposed to harmless colonizing bacteria along with food components essential to the development of tolerance, but it is also exposed to contaminants and pathogens which make the immune system able to fight efficiently against them [1,2]. The intestinal immune system can be divided into inductive and effector sites. Inductive sites include the gut-associated lymphoid tissues such as Peyer’s patches (PP), isolated lymphoid follicles and the gut-draining mesenteric lymph nodes (MLN). The *lamina propria* (LP) and the cellular epithelium constitute the main effector sites, harboring large populations of activated T cells and antibody-secreting plasma cells. Besides this, the intestinal antigen-presenting cells (APC), predominantly composed of macrophages and dendritic cells, play a central role in initiating and orchestrating immune responses. At homeostasis, they participate in the tolerance towards dietary components and colonizing commensal microbiota, but they also have the ability to fight pathogens [3]. In the presence of an infectious agent, APC are rapidly activated and contribute to the innate response by producing pro-inflammatory cytokines as well as presenting antigens to naive T cells, which can trigger a specific immune response. The drawback of this is that mucosal APC are present in a limited number and with low responsiveness in the intestinal mucosa of neonates.

Maternal diet supplementation with prebiotics during pregnancy improves the offspring immune defenses by supporting the structural development of the gut mucosa [4,5,6,7]. Herein, we selected short-chain fructooligosaccharides (scFOS), a highly interesting prebiotic fiber, since we obtained previously very interesting results after maternal and/or direct FOS supplementation on mucosal immune system development and functionality [5,8,9,10]. Direct dietary FOS supplementation also promotes intestinal IgA secretion [11,12]. In addition, previous studies demonstrated that maternal and/or direct FOS supplementation induced a better efficacy of vaccines against *Salmonella* in a mouse model [13], and against *Lawsonia* [10] and *Influenza* [9] in pigs. However, only a few studies have reported the impact of prebiotic supplementation on the number and functionality of APC in Peyer’s patches [14,15,16]. Moreover, modification in the microbiota composition and fermentative activity was observed in the offspring when the sow diet was supplemented with prebiotics [5,8].

In contrast, adverse events, such as environmental contamination, occurring in early life, may have negative consequences on adulthood [17,18,19,20]. Mycotoxins are secondary metabolites produced by fungi, which commonly contaminate human and animal foods. Given their global and frequent occurrence, their stability through the food-processing chain and their toxic effect, mycotoxins become a major concern in Europe [17,18,19]. Deoxynivalenol (DON) is a mycotoxin of the trichothecene family mainly produced by *Fusarium graminaerum* and *F. culmorum*. It has the ability to alter immune function [21,22] and particularly APC, which were identified in vitro as a target of mycotoxins [23,24]. Only a few in vivo studies investigated the impact of maternal diet contamination [25] and the transfer of DON across the placenta in pigs [26,27,28].

We hypothesized that different beneficial or deleterious dietary environments may induce changes in APC sub-populations throughout childhood and then participate in their susceptibility to infection. The objective of this study was to compare the impact of a beneficial maternal dietary scFOS supplementation vs. a detrimental DON contamination on the developmental profile of offspring immune system as a possible approach for improving animal health.

## 2. Materials and Methods

### 2.1. Animals, Diets and Experimental Design

The experimental protocol was designed in compliance with the legislation of the European Union (directive 86/609/EEC) and France (decree 2001-464 29/05/01) for the care and use of laboratory animals (agreement for animal housing number B-35-275-32 and authorization certificate # 006061 to experiment on live animals). The regional Ethics Committee in Animal Experiments approved the procedure described herein (R-2012-IL-02/07). Twelve sows (Large White × Landrace) and their piglets ((Large White × Landrace) × Pietrain) from the experimental unit physiology and phenotyping of pigs (UE3P, Saint-Gilles, France) were observed daily to ensure their welfare. Diets were formulated according to the nutrient and energy requirements of gestating and lactating sows (Cooperl, Lamballe, France). Feeds of gestating and lactating sows were supplemented with either maltodextrin as a control (named CTRL) (3.3 and 1.5 g/kg, respectively, Maldex, Tereos Starch & Sweeteners Europe, Marckolsheim, France, *n* = 4)) or scFOS prebiotic (named PREB) (3.3 and 1.5 g/kg, respectively, 95% of scFOS with molecular chain length between 3 and 5 monomeric unity, Profeed P95, Beghin-Meiji, Marckolsheim, France, *n* = 4) or deoxynivalenol (named DON) (3 mg/kg, Sigma, St. Quentin Fallavier, France, *n* = 4) (Appendix A). The CTRL sow group received the CTRL diets, while the PREB sow group received the scFOS diets, from the last 4th week of gestation and the entire lactation. Indeed, we previously observed that diet with such a low dose of scFOS during the gestation and lactation period was very effective in enhancing the gut immune system [5,10]. The DON sow group was fed the DON diet only for the last 4th week of gestation and the CTRL diet during lactation (Figure 1). Sows were given 3 kg/day of feed during gestation and fed ad libitum during lactation. Their feed intake was recorded weekly during the lactation. Sow body weight was recorded at 36 and 7 days before, and 14 and 28 days after parturition. Their back fat thickness was also measured ultrasonically (Sonolayer SAL-32B, Toshiba, Tokyo, Japan) at the P2-position on both sides of the sow 7 days before, and 14 and 28 days after parturition. In the 12 h following farrowing, the size of litters (number of stillborn and born alive piglets) and the individual piglet birth weights were recorded. When possible, the litter size was adjusted to 10–12 piglets by adding (when litter size < 10 piglets) or removing (when litter size > 12 piglets) piglets within each sows’ dietary group, without changing the mean litter birth weight. From one day (more or less 12 h) of age, piglets from CTRL sows (fed the CTRL diet during the last 4th week of gestation) were divided in two groups based on their birth weight and gender within each litter. Half of the piglets of each litter CTRL received by daily oro-gastric gavage a solution of DON (DONgav, 0.5 mg/kg/day, dissolved in dimethylsulfoxide), and the other half the excipient solution without DON (CTRL- negative control for PREB or DON diets) (Figure 1). The excipient was water with a quarter of blackcurrant syrup for the gavage of the dissolved DON or not. All piglets of DONgav and sham gavaged CTRL- groups were force-fed at the same time and up to 8 days of age (from PND1 to PND8). Piglets were weighed weekly and their mortality rate was recorded. Before weaning, sow-reared piglets had no access to creep feed. DONgav, CTRL-, PREB or DON, whose birth weights were close to the mean litter birth weight of their groups, were selected to be euthanized at PND10 (*n* = 4 per group) or PND21 (*n* = 4 per group) after 1 h of maternal separation. At weaning (PND28), piglets were fed ad libitum a commercial starter 1st phase diet for two weeks, then a commercial starter 2nd phase for two weeks and finally a commercial growing diet (Cooperl) (Appendix A). They were weighed weekly. Finally, piglets from DONgav, CTRL-, PREB or DON groups were euthanized at PND49 (*n* = 4 per group) and only piglets from CTRL- and PREB were also euthanized at PND90 (*n* = 4 per group) after 1 h fasting (Figure 1).

### 2.2. Immune Cell Dissociation from Jejunum and Ileum

At slaughter, after opening the intestinal cavity, a 10 cm segment of jejunum containing discrete Peyer’s patches (jej-PP) or not (*lamina propria*, jej-LP) and a 10 cm segment of distal ileum with continuous Peyer’s patches (il-PP) were collected (Figure 1) and rinsed with ice-cold calcium and magnesium-free Hank’s balanced salt solution (HBSS, Sigma) supplemented with 50 µg/mL gentamicin (Sigma), 200 UI/mL penicillin, 200 µg/mL streptomycin (Sigma) and 10 mM HEPES (Sigma). Mononuclear immune cells from jej-LP, jej-PP and il-PP were purified as detailed [5]. Briefly, pieces of jejunum (jej-PP and jej-LP) and ileum (il-PP) were incubated twice for 45 min in HBSS, containing 1 mM of ethylenediaminetetra-acetic acid (EDTA) at 37 °C on a shaking platform. Cells were suspended in a 40% Percoll in RPMI solution and underlaid with an 80% Percoll solution (Pharmacia, Uppsala, Sweden). Low-density cells were recovered from the 40% to 80% gradient interface. The impact of prebiotic (CTRL- vs. PREB) was analyzed on mononuclear immune cells extracted from jej-PP at PND49 and PND90 and from il-PP at PND10, PND21, PND49 and PND90. We used immune cells from il-PP and jej-PP at PND90, as we recognized the effects of perinatal maternal scFOS on offspring immune system, long after his birth [8,10]. Previous studies have demonstrated that the intestinal absorption of DON takes mainly place in the jejunum directly from the intestinal lumen to the apical side of the intestinal epithelium [29]. Therefore, to analyze the influence of DON (group CTRL-, DONgav and DON), we used immune cells purified from jej-LP at PND10, PND21 and PND49. PP and LP immune cells were suspended in CellWash (BD Biosciences, le Pont de Claix, France) for flow cytometry analysis.

### 2.3. Flow Cytometry Analysis

Flow cytometry analysis was performed on the APC of PP and LP immune cell samples using specific monoclonal antibodies (mAbs) or isotype-matched mAb (as controls). Cells were incubated with primary mouse mAbs (20 min, 4 °C) recognizing porcine cell surface antigens: MHC class II (MHCII, SLA-DQ; clone MSA3, mouse IgG2a, Washington State University Monoclonal Antibody Centre, Pullman, USA) with CD16 (clone G7, mouse IgG1) or CD11R1 (CD11b; clone MIL4, mouse IgG1), both purchased from Bio-Rad (AbD Serotec, Oxford, UK). After two washes in CellWash, PP and LP, cells were finally incubated (20 min, 4 °C) with fluorochrome conjugated goat anti-mouse immunoglobulin isotype antibodies (anti-mouse IgG2a-FITC for MHCII, and mouse IgG1-RPE for CD11R1 or CD16, Clinisciences, Nanterre, France). Cells were analyzed with a MACSQuant analyser (Miltenyi Biotech, Paris, France) equipped with MACSQuantify software in order to evaluate the proportion of the APC sub-populations CD16+ MHCII+ (resident or non-migrant population) and CD11R1+ MHCII+ (migrant population from mucosa to the MLN) [30].

### 2.4. Explant Cultures

Intestinal explants were used to test the reactivity of the mucosal immune system [31,32]. Adapted from Chatelais et al. [33], 5 cm segments of jej-PP, jej-LP and il-PP were rinsed with PBS containing 1% dithiothreitol (DTT, Sigma) and 1% FCS (Sigma) and then placed in a solution made with 74% PBS, 25% DMEM (Sigma), 1% FCS, 5 µg/mL gentamicin (Sigma) and 1.25 µg/mL amphotericin B (Sigma) for immediate explant culture. Jejunal and ileal mucosa were then cut in small biopsies of 10–15 mg. Biopsies (two per well, weighing, in total, 20 to 35 mg) were cultured for 2 h under an atmosphere containing 5% CO_2_ at 37 °C [33]. Explants were then transferred into the same solution as above but without FCS (replacement with BSA), and stimulated or not with 50 µg/mL lipopolysaccharides (LPS, TLR4 ligand; Sigma) or 1 µg/mL flagellin (TLR5 ligand; Invivogen, Toulouse, France) and incubated for 20 h under an atmosphere containing 5% CO_2_ at 37 °C. Finally, supernatants were collected and frozen at −20 °C for later cytokine analysis, and the biopsies were harvested in 1 mL of Trizol reagent (Fisher Scientific, Illkirch, France) and frozen at −80 °C for RNA extraction.

### 2.5. Determination of Cytokine Concentration

Cytokines (IL-1β, IL-6, IL-8, TNFα and IL-10; Appendix A) were measured in the supernatant of explant cultures. Concentrations of porcine cytokines were determined by ELISA (R&D Systems Europe, Lilles, France). In basal condition (without stimulation), values were expressed as pg/mg of tissue, and following TLR stimulation, secreted cytokines were expressed as a ratio between TLR stimulated condition and basal condition.

### 2.6. RNA Extraction and Quantitative Real-Time Polymerase Chain Reaction (qRT-PCR) Analysis

The biopsies (from basal and stimulated conditions) were thawed and lysed with 1 mL of Trizol reagent with the Precellys tissue homogenizer (Bertin technologies, Rockville, CA, USA). Total RNA was purified using RNeasy Mini Kit (Qiagen, Courtaboeuf, France) and residual genomic DNA was removed using DNase digestion with RNase-free DNase I Amp Grade (Fisher Scientific). RNA amount was quantified using a NanoDrop spectrophotometer and Agilent 2100 Bioanalyser (Agilent Technologies France, Massy, France) monitored its integrity. RNA samples (2 g) were reverse-transcribed using High Capacity cDNA-RT kit (Thermo Fisher Scientific) [8]. Quantitative RT-PCR was performed in 384-well plates in a ViiA7 thermocycler (Thermo Fisher Scientific), as already described [21,34]. Primers for the qRT-PCR were presented in Appendix A [35,36,37]. Cyclophilin A and RPL32 genes were used as housekeeping genes as they exhibit high-stability values among samples and were not significantly affected by dietary treatments. The relative gene expression was determined using the cycle threshold (Ct) method and normalized to the expression of the two housekeeping genes (Ct of gene of interest over mean Ct). The relative expressions of the target genes were determined using the 2^−ΔΔCt^ method.

### 2.7. Statistical Analysis

The data were assessed for normality using D’Agostino and Pearson omnibus K2 normality test and analyzed using the GraphPad Prism (GraphPad Software, version 6.01, San Diego, CA, USA). Normally distributed data were analyzed using a two-way ANOVA performing time, diet and the interaction between these two factors, followed by Tukey’s multiple comparison tests when appropriate (data on sow and piglet performances, Figure 2 and Figure 3). Non-parametric datasets (flow cytometry, qRT-PCR and ELISA data) were analyzed using the Mann-Whitney test (CTRL- vs. PREB, Figure 4, Figure 5 and Figure 6) or the Kruskal-Wallis test followed by Dunn’s multiple comparison tests when appropriate (CTRL- vs. DONgav vs. DON Figure 7, Figure 8 and Figure 9). The results were presented as mean ± SEM (sow and piglet performances) or mean ± SD (flow cytometry, qRT-PCR and ELISA data). We made the statistical analysis at each time point but we presented only the statistically differences or tendencies. Different letters represented significant differences between different time groups, stars (significant differences at *p* ≤ 0.05) and hashes (tendencies when 0.05 < *p* ≤ 0.10) represented differences between different treatment groups.

## 3. Results

### 3.1. Sow Diet. Prebiotic Supplementation or Mycotoxin Contamination on Sow Performance during Gestation and Lactation

Body weight of CTRL, PREB and DON sows was significantly increased during gestation (time effect: D- 36 vs. D- 7, *p* < 0.05) and decreased after farrowing (time effect: D- 7 vs. D+ 14 or vs. D+ 28, *p* < 0.05) (Figure 2A,D). PREB supplementation had a positive impact since PREB sows at D+ 14 and D+ 28 kept quite constant their back fat thickness compared to CTRL sows (* *p* < 0.05, Figure 2B). However, the back fat thickness of DON and CTRL groups significantly decreased from one week before farrowing (time effect: D- 7 vs. D+ 14 and D- 7 vs. D+ 28 for CTRL and DON, *p* < 0.05) (Figure 2E). The dietary intake of CTRL and PREB sows, on the one hand and CTRL and DON sows, on the other hand, was similarly increased with lactation (time effect vs. D+ 7, *p* < 0.05, Figure 2C,F).

### 3.2. Sow Diet. Prebiotic Supplementation or Mycotoxin Contamination Differentially Influenced Piglet Performance

Maternal diet supplementation with PREB or contamination with DON did not affect litter size and piglet birth weight, nor mortality rate, during the first two days of life (Appendix A). However, 37% ± 16 of the piglets that received DON by oral gavage during their first 8 days of life (DONgav, *p* < 0.06) died between PND3 and PND9, while no death was recorded in the two other groups during this period (Figure 3A). The piglets that were force-fed with DON demonstrated a lower growth rate as soon as PND7, resulting in a lower body weight compared to CTRL- and DONgav piglets (*p* < 0.0001) (Figure 3B). CTRL- and PREB pigs grown normally from birth to PND49 without any differences between groups (Figure 3C).

CTRL- and PREB groups showed an increased mononuclear cell recruitment in il-PP from PND21 until PND49 (Figure 4A) and a decrease from PND49 to PND90, without any differences between groups. The recruitment of immune cells in the jej-PP did not change significantly between PND49 and PND90, or between groups (data not shown). Maternal supplementation with PREB did not affect the proportion of CD16+ MHCII+ and CD11R1+ MHCII+ APC in il-PP and jej-PP (Figure 4B–E).

### 3.3. Sow Diet. Prebiotic Supplementation Changed Gene Expression and Cytokine Patterns of Unstimulated il-PP Explants

Sow diet prebiotic supplementation did not change the gene expression and cytokine patterns of jej-PP in unstimulated condition, nor after TLR stimulation (data not shown). In contrast, unstimulated PREB il-PP explants displayed down-regulation of TGFβ (*p* < 0.06) and IL-23A (*p* < 0.03) gene expression at PND10, up-regulation of TLR4 gene expression (*p* < 0.03) at PND21, and down-regulation of IL-1β (*p* < 0.10) and TLR5 (*p* < 0.03) gene expression as well as a tendency for an up-regulation of IL-12p40 (*p* < 0.06) gene expression at PND90, compared to CTRL- il-PP explants (Figure 5A). Regarding cytokine secretions, IL-8 production was higher at PND10 (*p* < 0.06), whereas IL-1β (*p* < 0.03) and IL-10 (*p* < 0.06) secretions were lower at PND90 in PREB il-PP explants compared to CTRL- il-PP explants (Figure 5B). The il-PP explants obviously contained several cells and each of them could secrete different cytokines, even if they have opposite effects. The different cells (macrophages, Treg and Th2 CD4+ lymphocyte cells) produce opposite cytokines (IL-1β and IL-10) which could be affected differentially in il-PP explants, even in basal conditions.

### 3.4. Sow Diet. Prebiotic Supplementation Changed Gene Expression and Cytokine Patterns of il-PP Explants Stimulated with TLR-ligands

We also tested the gene expression and cytokine secretion of il-PP explants stimulated with 1 µg/mL flagellin or with 50 µg/mL LPS. At PND10, stimulated PREB il-PP explants displayed increased gene expression of TGFβ (*p* < 0.06 with flagellin), IL-23A (*p* < 0.03 with flagellin and LPS) and FoxP3 (*p* < 0.06 with LPS) compared to CTRL- il-PP explants. At PND21, IL-10 (*p* < 0.06) and CX3CL1 (*p* < 0.06) gene expression tended to be down-regulated in flagellin-stimulated PREB il-PP explants compared to flagellin-stimulated CTRL- il-PP explants. At PND90, TLR5 gene expression tended to be up-regulated (*p* < 0.06) in LPS-stimulated PREB il-PP explants compared to LPS-stimulated CTRL- il-PP explants (Figure 6A). Considering cytokine secretions, at PND49, LPS-stimulated PREB il-PP explants tended to secrete more TNFα (*p* < 0.06) and IL-10 (*p* = 0.10) compared to LPS-stimulated CTRL- il-PP explants. As for immune cells from unstimulated explants, the different cells in LPS-stimulated explants (neutrophils, macrophages, NK cells for TNFα or Treg and Th2 CD4+ lymphocyte cells for IL-10) could be affected differentially in PREB il-PP explants. At PND90, stimulated PREB il-PP explants tended to secrete less IL-1β (*p* < 0.06) in response to flagellin compared to stimulated CTRL- il-PP explants (Figure 6B).

### 3.5. Sow Diet. Mycotoxin Contamination (DON) and Neonate Mycotoxin Gavage (DONgav) Modified jej-LP Cell Numbers and the Proportion of Resident APC

Maternal exposure to DON (DON), but not DON gavage of piglets during the first week of life (DONgav), tended to increase (# *p* < 0.10) immune cell recruitment and/or local proliferation in piglet jej-LP at PND21 compared to CTRL- (Figure 7A). Maternal diet DON contamination (DON) and DON gavage of piglets (DONgav) decreased the proportion of resident CD16+ MHCII+ APC, but not that of CD11R1+ MHCII+ APC, in jej-LP (Figure 7B,C). The proportion of CD16+ MHCII+ APC was 3.5-fold reduced (*p* < 0.02) in DON compared to CTRL- jej-LP at PND10, while it was 4-fold reduced (*p* < 0.02) in DONgav compared to CTRL- jej-LP at PND21.

### 3.6. Sow Diet. Mycotoxin Contamination (DON) and Neonate Mycotoxin Gavage (DONgav) Changed Some Gene Expression of both Unstimulated and Stimulated jej-LP Explants

Basically, the recent study by Alizadeh and colleagues summarized the negative impacts of DON on the villus architecture and expression levels of genes related to intestinal barrier function, oxidative stress and inflammation which were mainly observed in the jejunum compared to other parts of intestine [38]. Maternal diet mycotoxin exposure (DON) increased (* *p* < 0.05) IFNγ gene expression in comparison with neonatal gavage with DON (DONgav) and vehicle (CTRL-). BAFF gene expression has a tendency to be reduced in DONgav unstimulated jej-LP explants at PND10 (Figure 8). In contrast, at PND49, BAFF gene expression tended to be higher in DONgav compared to CTRL- and DON unstimulated jej-LP explants (Figure 8).

CX3CL1, IL-12p40, IL-10, BAFF and IL-1β gene expressions at PND10, and FoxP3 and CCL20 gene expressions at PND49 were modified in jej-LP explants stimulated with flagellin or LPS (Figure 9). At PND10, flagellin-stimulated DONgav jej-LP explants tended to display higher CX3CL1 (*p* < 0.06) gene expression compared to DON, and lower IL-12p40 (*p* < 0.06) gene expression compared to CTRL- jej-LP explants. At PND49, flagellin-stimulated DONgav jej-LP explants tended to display lower FoxP3 (*p* < 0.10) gene expression compared to CTRL- jej-LP explants, while flagellin- and LPS-stimulated DONgav jej-LP explants displayed lower CCL20 (*p* < 0.04 and *p* < 0.10, respectively) gene expression compared to DON jej-LP explants (Figure 9).

## 4. Discussion

Our study aimed to compare the impact of a beneficial vs. a detrimental maternal dietary environment on their offspring immune system developmental profiles during the neonatal period. Maternal prebiotic supplementation or DON contamination diet did not change the profile of body weight variations in sow during gestation and lactation compared to CTRL sows. Nevertheless, the maternal prebiotic diet contributed to save up sow body reserves. The consequence of sow diet DON contamination on offspring piglet growth appeared only after weaning. In contrast, early piglet gavage with DON induced either death or a major growth reduction for surviving piglets that became apparent as soon as two weeks of life. This illustrated the ability of the placental barrier to preserve, at least partly, fetuses from mycotoxin contamination, as previously observed [26]. The toxicity of DON varied according to several parameters such as the dose, the duration of exposure, the animal age as well as nutritional factors [39]. Previous studies reported negative impacts on the performance and gut hormone secretion of naturally mycotoxin-contaminated feed for piglets (1 to 3 mg/kg) during a 21 day-period [40]. In addition, gut health and integrity were affected in 6-week-old growing pigs with a low dose (0.9 mg/kg feed) for 10 days [38]. Thus, the direct consumption of mycotoxin-contaminated feed by piglets had more detrimental/harmful impacts on offspring health than indirectly via maternal feed contamination.

We intended to gain insight into how maternal perinatal prebiotic supplementation or mycotoxin contamination affected the developmental profile of offspring immune system, and we decided to focus on the intestinal responsiveness of gut immunity during the first three months of piglet life. Previous observations in growing female rats suggested that prebiotic (inulin, long-chain FOS) dietary supplementation increased the proportion and the number of dendritic cells in Peyer’s patches [15]. In our study, maternal prebiotic supplementation did not affect the proportion of the APC sub-populations CD16+ MHCII+ (non-migrating APC) and CD11R1+ MHCII+ (migrant APC of the mucosa towards MLN) in both il-PP and jej-PP. The difference may be attributed to the different animal species, the mode of prebiotic supplementation (direct vs. maternal supplementation), the source and more specifically the dose of prebiotics tested (5% in rats compared to 0.15–0.33% in our study). Using specific markers for macrophages (CD14) and the sub-populations of dendritic cells (CD172α, signal regulatory protein, SIRPα) would help to specifically distinguish APC sub-populations.

It is important to note that maternal prebiotic supplementation did not change any gene expression and cytokine secretion in offspring jej-PP. Unstimulated il-PP explants from PREB piglets displayed lower TGFβ and IL-23A gene expression at PND10, associated with higher secretion of pro-inflammatory IL-8 cytokine, whereas after TLR ligand (flagellin or LPS) stimulation, il-PP explants expressed higher TGFβ, IL-23A and FoxP3 gene levels. These markers are involved in tolerogenic responses by FoxP3+ regulatory T (Treg) cells, a process requiring TGFβ as a key modulator of gut homeostasis and tolerance [41]. The increased IL-23 gene expression observed in TLR-stimulated conditions could orchestrate tolerogenic environment, and thus the development of FoxP3+ Treg-immune cells [42]. This statement should be confirmed by a specific membrane and intracellular staining of CD4+ CD25+ FoxP3+ T cells by flow cytometry. At PND21, unstimulated PREB il-PP explants exhibited a significant up-regulation of TLR4, suggesting a possible impact of prebiotics on epithelial cells. Indeed, the immune response was markedly decreased by TLR4 gene knock-down, highlighting that prebiotics, including scFOS, could be TLR4-ligand on intestinal epithelial cells [43,44]. The up-regulation of TLR5 gene expression observed in PREB il-PP explants, after stimulation with LPS, corroborated the higher expression of TLR5 reported in five-week-old mice PP after an oral probiotic administration [45], suggesting a main role of gut bacteria in the control of TLR expression, and by extension, the control of the immune system development. The down-regulation of CX3CL1 gene expression in PREB il-PP explants, after stimulation with flagellin, could predestine a reduced migration of cells in il-PP explants at PND21 [46,47]. Finally, the pro-inflammatory IL-1β gene expression and secretion decreased at PND90 with and without TLR activation. Overall, maternal supplementation with prebiotic diet during perinatal period did not influence the ontogenesis of APC in offspring il-PP and jej-PP. However, the mucosal immune system of offspring born from prebiotic-supplemented diet sows early displayed higher inflammatory cytokine IL-8 gene expression when unstimulated, and further promoted T regulatory cells after TLR activation with decline of IL-1β level.

One of the effects of maternal DON exposure compared to DON force-fed neonates and sham force-fed control neonates was an impact on the offspring jej-LP mononuclear cell number. Immune cell recruitment, at PND21, was significantly increased in the offspring of DON-contaminated sows compared to both other groups. We could assume the transfer of the DON from the mother to the offspring via colostrum [28]. However, maternal transfer has less impact than direct exposure by gavage of DON on offspring growth, as illustrated by our results. APC were described in vitro [29,48] and in vivo [48] as a target of mycotoxins. DON contamination of maternal diet decreased the proportion of the APC sub-population CD16+ MHCII+, but not one of the CD11R1+ MHCII+ into jej-LP immune cells of offspring as soon as PND10. DON gavage to piglets, immediately after birth, led also to a decrease in the same APC subset CD16+ MHCII+ at PND21. These results contrasted with the expansion of the CD16+ immune cells in the epithelium of jejunum of fattening pigs with DON-contaminated feed at the same dose of DON [49]. However, other cell subsets can express CD16, such as NK cells [50]. Overall, the adverse effects of DON were always greater in young animals (3- to 4-week-old) compared to growing animals (8 to 10-week-old) [51]. It was previously reported that DON increase the number of CD16+ cells migrating from the *lamina propria* into the epithelium of the jejunum [49]. This result can explain the decrease in the resident CD16+ MHCII+ APC observed in our work. Moreover, the changes in the laminin production and in epithelium composition by DON can provoke an increase in the pore number in the jejunum, which is the basis for an increased migration of CD16+ cells into the epithelium. Diesing et al. [52] showed that high DON concentrations on the enterocyte border induced cell death and loss of the epithelial barrier integrity. In tandem with this result, recent work of Vignal et al. showed, in a mice model, that DON-induced dysbiosis with an enterobacterial bloom could contribute to DON effects on intestinal inflammation [53]. This study revealed the decrease in Bacteroidetes and the increase in Proteobacteria in DON-exposed mice compared to control mice.

Unstimulated jej-LP explants of piglets from DON sows showed a significantly higher gene expression for IFNγ at PND10 demonstrating that maternal exposure to this mycotoxin can produce conditions for polarization of Th1 responses increasing piglet protection against intracellular pathogen infection. At PND49, we observed for DONgav piglets a trend for higher expression of BAFF gene compared to the other groups. A similar pattern of BAFF gene expression was observed following LPS stimulation at PND10, suggesting a shift towards humoral immunity. Indeed, previous research has shown that DON may increase total serum IgA levels [54]. Nevertheless, IgA production by stimulating B cells through BAFF requires the involvement of APRIL, TGFβ and antigen-containing immuno-complexes [55]. Interestingly, at PND10, we noticed an up-regulation of IL-1β gene expression in the presence of LPS in DONgav and maternal DON-exposure piglets compared to CTRL- piglets, demonstrating a pro-inflammatory response. Previous works showed also that, in mice, simultaneous exposure to sub-toxic intravenous doses of LPS and dietary DON caused a sequential elevation of IL-1β over-expression [56]. We also observed in the intestinal explants of piglets from DON sows at PND49, but not with the one of force-fed piglets, that the CCL20 gene was one of the most up-regulated genes [49]. CCL20 is implicated in the formation and function of mucosal lymphoid tissues via chemo-attraction of lymphocytes and dendritic cells towards the epithelial cells. Then, DON contamination of sow and piglets decreased CD16+ MHCII+ APC at PND10 in *lamina propria* associated with IFN inflammation and impairment of Treg response all the more than FoxP3+ gene expression seemed to decrease in DONgav il-LP explants at PND49. In the intestine, resident dendritic cells play a crucial role in tolerogenic responses including regulatory T cell induction [57]. DON influences the epithelial cell turnover in the small intestine [53]. Exposure to DON also induced inhibition of the co-stimulatory factor CD86 in resident dendritic cells. This can be attributed to regulatory T cells, which are able to inhibit CD86 expression in dendritic cells, which attenuates their maturation and stimulatory function [58].

The findings of this study should be interpreted in light of its inherent limitations. Indeed, the piglets of the CTRL- group were force-fed (sham) with excipient from 1 to 8 days of age at the same time as the piglets of DONgav group. Therefore, the significant stress that may have been induced by the gavage of animals must be taken into account when comparing CTRL- and PREB groups as well as CTRL- and DON groups.

## 5. Conclusions

Herein, we demonstrated for the first time that perinatal exposure to DON directly by gavage or to a lower extent, via supplementation of mothers, impacted the proportion of *lamina propria* APC leading to a mucosal immune system impairment of the Treg response. In contrast, maternal supplementation with scFOS did not affect the proportion of offspring il-PP APC, but influenced the responsiveness of gut mucosal immunity by the generation of Treg response after TLR stimulation. It would be interesting to extend this study by evaluating the preventing effects of maternal prebiotic supplementation on the adverse effects of DON contamination on the mucosal immune system developmental profile, as suggested in the study of Ferrer et al. [59].

## Figures and Tables

**Figure 1 nutrients-12-02115-f001:**
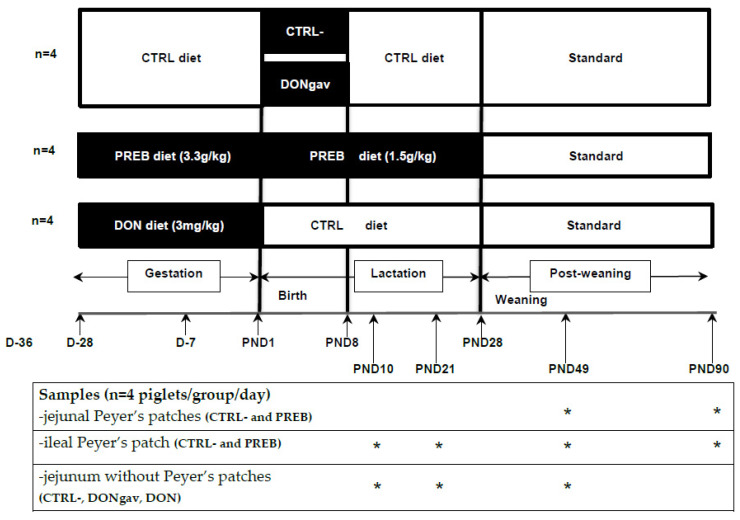
Experimental design. Twenty-eight days before the expected time of farrowing and until the end of the lactation (PND28), two groups of sows were fed either a maltodextrin diet as a control (CTRL group, *n* = 4) or a scFOS-supplemented diet (PREB group, *n* = 4). The third group of sows were fed with DON-contaminated diet (DON group, *n* = 4) during the gestation period only. Piglets from PREB and DON sow litters were selected for PREB and DON piglet groups, respectively. Within the CTRL group litter, piglet received daily a DON solution (DONgav) or excipient (CTRL-, negative control for PREB and DON diets) from PND1 to PND8 orally. Piglets were euthanized at PND10, PND21, PND49 or PND90. The stars in the table represented the PND when jejunal with or without Peyer’s patches and ileal Peyer’s patches were analyze.

**Figure 2 nutrients-12-02115-f002:**
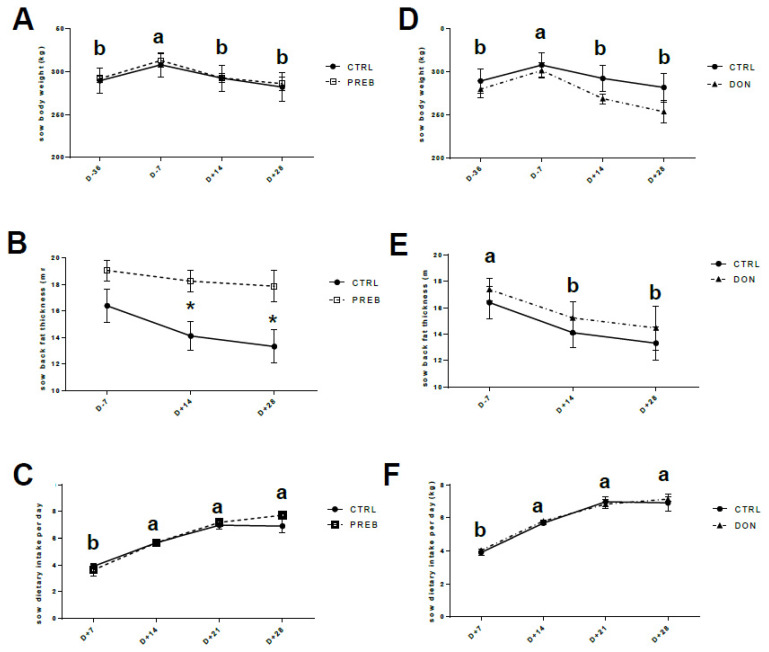
Sow performance. On left, comparison of control (*n* = 4, CTRL, full circle) and PREB (*n* = 4, PREB, empty square) groups and on right, comparison of CTRL and contaminated sows with DON (*n* = 4, DON, full triangle). Normally distributed data were analyzed using a two-way ANOVA performing time, diet and the interaction between these two factors, followed by Tukey’s multiple comparison tests when appropriate Time effect: a, b means with different letters are significantly different (*p* < 0.05). (**A**,**D**) Sow body weight (kg) was recorded at 36 and 7 days before, and 14 and 28 days after parturition. (**B**,**E**) Back fat thickness (mm) was measured ultrasonically at 7 days before, and 14 and 28 days after parturition (significant difference at (* *p* < 0.05) for CTRL and PREB). (**C**,**F**) Sow feed intake (kg) was recorded weekly from D+ 7 to D+ 28 after parturition.

**Figure 3 nutrients-12-02115-f003:**
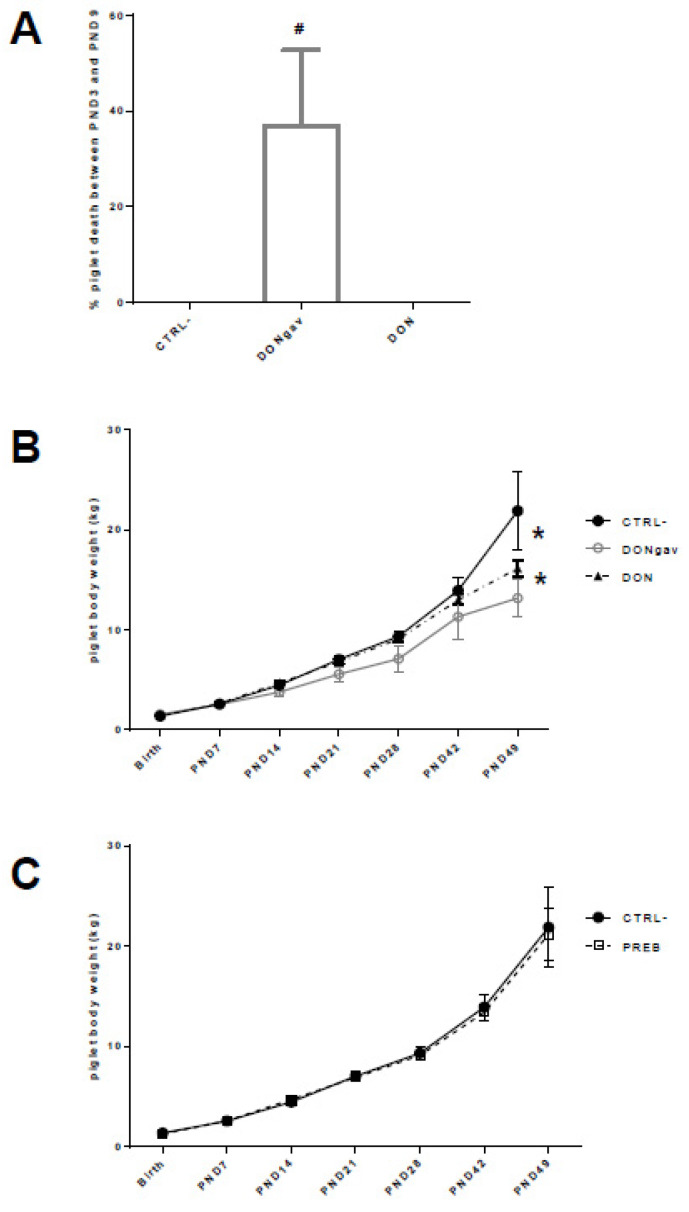
Piglet performance. Normally distributed data were analyzed using a two-way ANOVA performing time, diet and the interaction between these two factors, followed by Tukey’s multiple comparison tests when appropriate (**A**) Piglet death proportion (% of the litter size) from PND3 to PND9 after force-fed gavage with DON (DONgav) compared to control (CTRL-) and maternal supplementation with DON (DON). # means difference at *p* = 0.06 between DONgav and the two other groups (CTRL- and DON). (**B**) Piglet body weight (kg) from birth to PND49. Time, diet and the interaction time x diet were highly significant (*p* < 0.0001). DONgav piglet body weights (empty grey circle) were lower than CTRL- (full black circle) and DON piglets (full black triangle) (* *p* < 0.0001). (**C**) Body weights of piglets CTRL- (full black circle) and PREB (empty black square) recorded from birth to PND49 were not different.

**Figure 4 nutrients-12-02115-f004:**
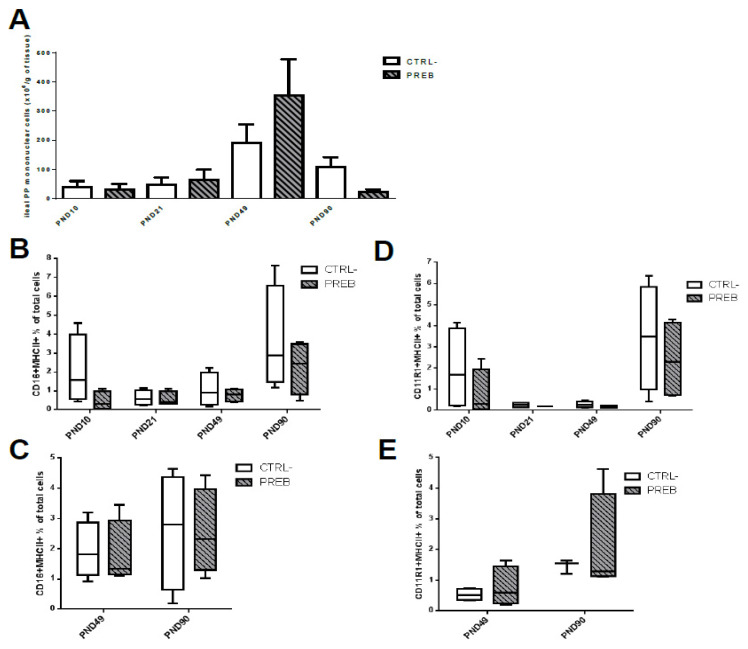
Sow diet prebiotic supplementation impacted ileal and jejunal PP cell numbers, but not the proportion of resident and migrating APC. Datasets were analyzed using the non-parametric Mann-Whitney test, for each age. (**A**) Numeration of mononuclear immune cells (×10^6^ per g tissue) in il-PP of CTRL- (*n* = 4, empty black bar) and PREB (*n* = 4, black hatched grey bar) piglets at PND10, PND21, PND49 and PND90. (**B**) CD16+ MHCII+ proportion (% of total immune cells measured by flow cytometry) in CTRL- and PREB il-PP at PND10, PND21, PND49 and PND90. (**C**) CD16+ MHCII+ proportion (% of total immune cells) in CTRL- (*n* = 4, empty black bar) and PREB (*n* = 4, black hatched grey) jej-PP at PND49 and PND90. (**D**) CD11R1+ MHCII+ proportion (% of total immune cells) in CTRL- (*n* = 4, empty black bar) and PREB (*n* = 4, black hatched grey bar) il-PP at PND10, PND21, PND49 and PND90. (**E**) CD11R1+ MHCII+ proportion (% of total immune cells) in CTRL- (*n* = 4, empty black bar) and PREB (*n* = 4, black hatched grey bar) jej-PP at PND49 and PND90.

**Figure 5 nutrients-12-02115-f005:**
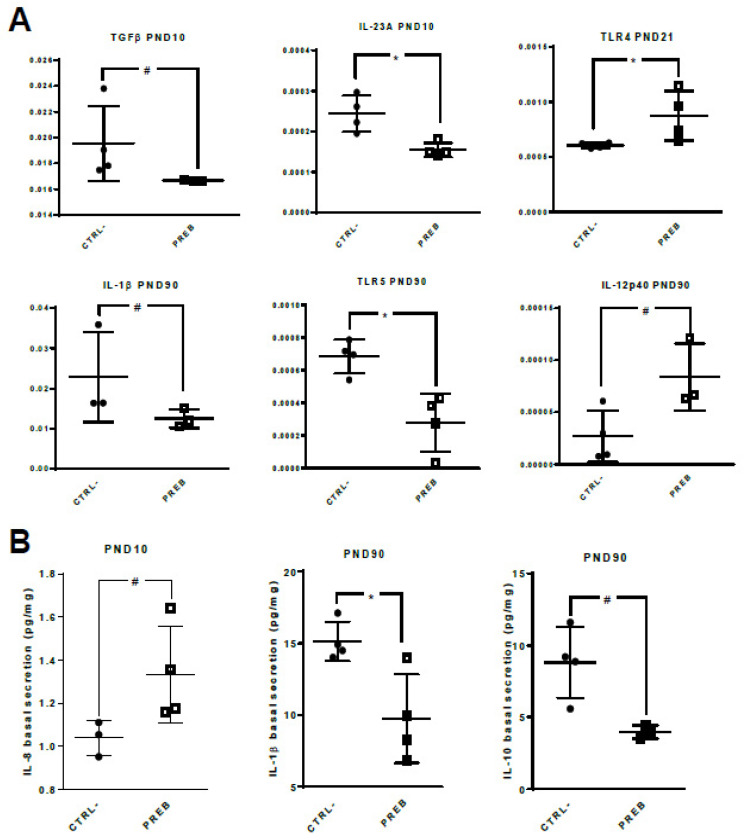
Sow diet prebiotic supplementation effect on gene expression and cytokine patterns of unstimulated il-PP explants. Datasets were analyzed using the non-parametric Mann-Whitney test. (**A**) qRT-PCR gene expression from unstimulated il-PP explants of CTRL- (*n* = 3–4, full circle) and PREB (*n* = 3–4, empty square) piglets. The relative gene expression was determined using the cycle threshold (Ct) method and normalized to the expression of the two housekeeping genes (Ct of gene of interest over mean Ct). Differences in il-PP explant gene expression levels between CTRL- vs. PREB are displayed with * *p* < 0.05, # *p* < 0.10. (**B**) Cytokine secretion in unstimulated il-PP explant from CTRL- (*n* = 3–4, full circle) and PREB (*n* = 3–4, empty square) piglets. Differences in il-PP explant cytokine secretion between CTRL- vs. PREB are displayed with * *p* < 0.05, # *p* < 0.10.

**Figure 6 nutrients-12-02115-f006:**
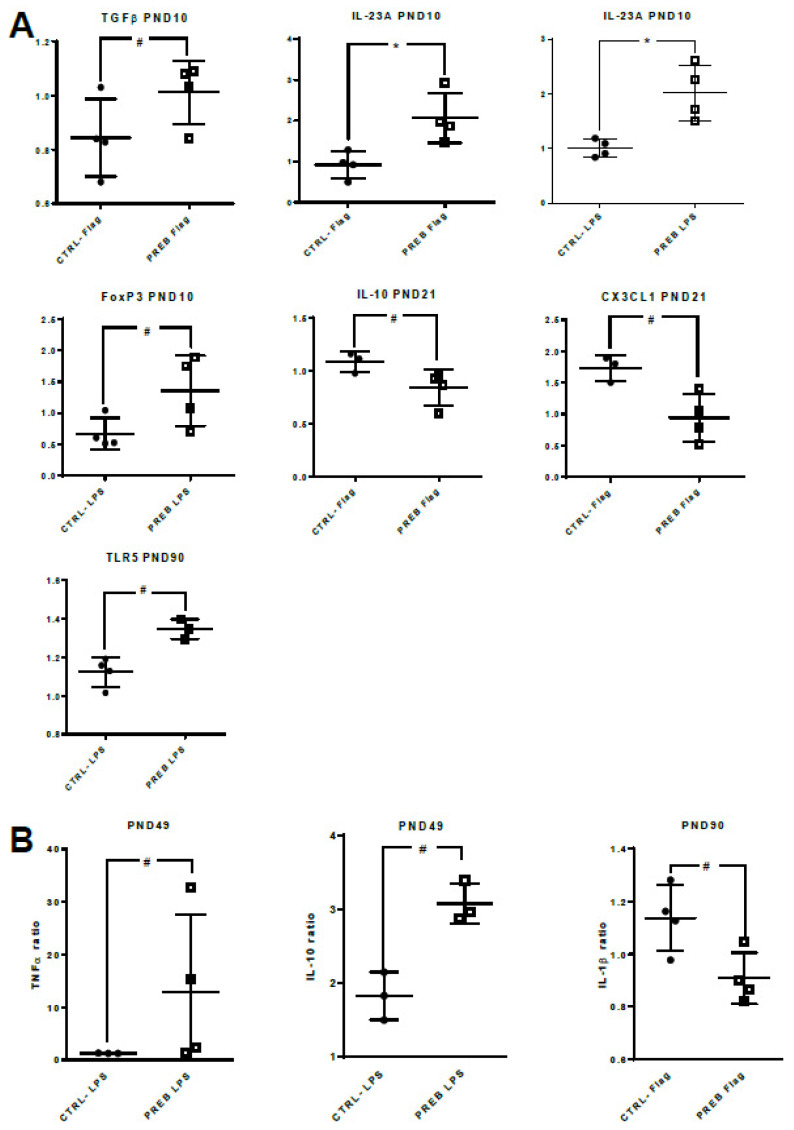
Sow diet prebiotic supplementation effect on gene expression and cytokine patterns of il-PP explants stimulated with TLR-ligands. Datasets were analyzed using the non-parametric Mann-Whitney test. (**A**) Relative expressions of the target genes were determined using the 2^−ΔΔCt^ method. Il-PP explant gene expressions after stimulation with LPS (50 µg/mL) or with flagellin (1 µg/mL, Flag) between CTRL- (*n* = 3–4, full circle) and PREB (*n* = 3–4, empty square) piglets (* *p* < 0.05, # *p* < 0.10). (**B**) Il-PP explant cytokine secretion following TLR-ligand stimulation between CTRL- (*n* = 3–4, full circle) and PREB (*n* = 3–4, empty square) (# *p* < 0.10).

**Figure 7 nutrients-12-02115-f007:**
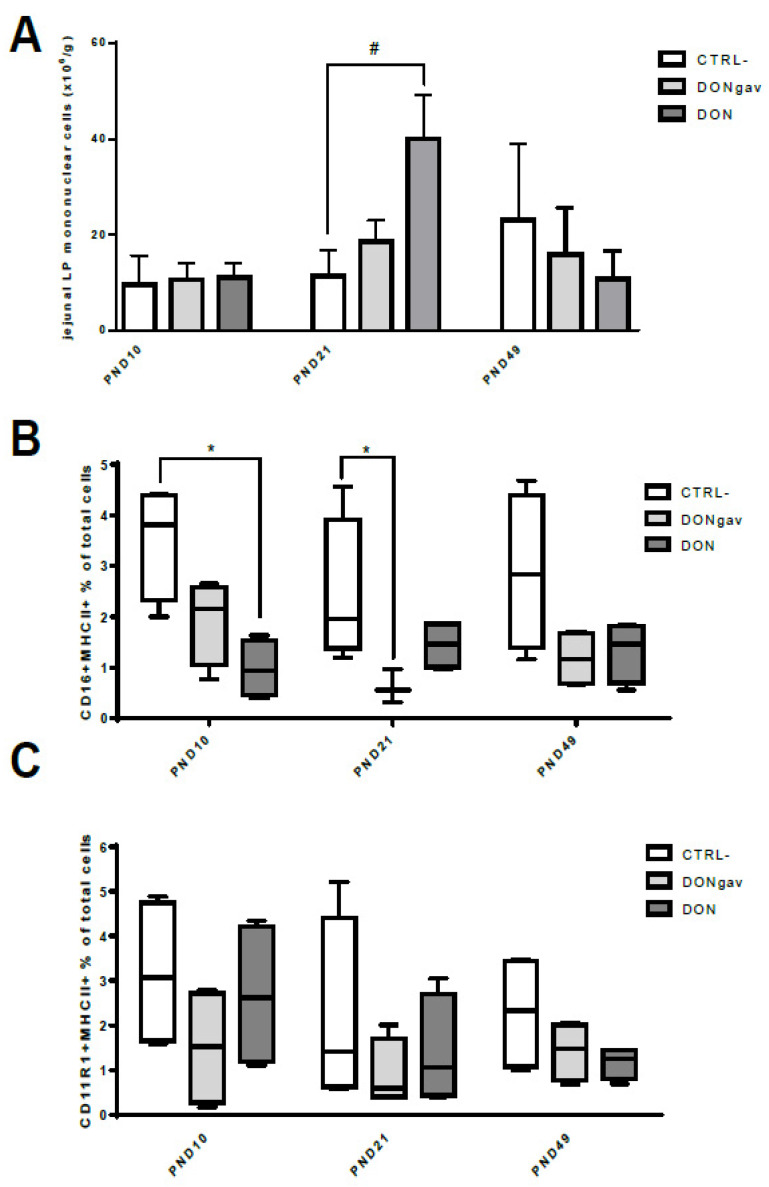
Sow diet mycotoxin contamination (DON) and neonate mycotoxin gavage (DONgav) modified jej-LP cell numbers and the proportion of resident APC. Datasets were analyzed with the non-parametric Kruskal-Wallis test followed by Dunn’s multiple comparison tests when appropriate for each age. (**A**) Numeration of immune mononuclear cells (× 10^6^ per g of tissue) isolated from jej-LP at PND10, PND21 and PND49 from CTRL- (*n* = 4, empty black bar), DONgav (*n* = 4, light grey bar) and DON (*n* = 4, dark grey bar) piglets (# *p* < 0.10). (**B**) CD16+ MHCII+ proportion (% of total immune cells) in jej-LP CTRL- (*n* = 4, empty black bar), DONgav (*n* = 4, light grey bar), and DON (*n* = 4, dark grey bar) explants (* *p* < 0.02). (**C**) CD11R1+ MHCII+ proportion (% of total immune cells) in jej-LP CTRL- (*n* = 4, empty black bar), DONgav (*n* = 4, light grey bar) and DON (*n* = 4, dark grey bar).

**Figure 8 nutrients-12-02115-f008:**
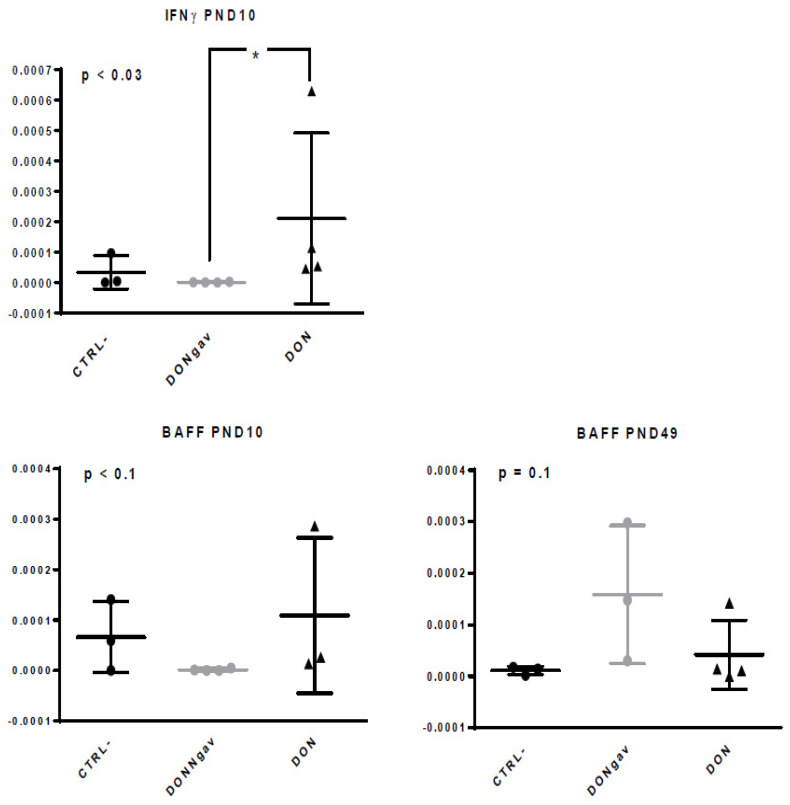
Sow diet mycotoxin contamination (DON) and neonate mycotoxin gavage (DONgav) effect on gene expression of unstimulated jej-LP explants. Datasets were analyzed with the non-parametric Kruskal-Wallis test (*p*-value on left corner of each graph) followed by Dunn’s multiple comparison tests when appropriate (CTRL- vs. DONgav vs. DON). qRT-PCR gene expression of unstimulated jej-LP explants of CTRL- (*n* = 3, full black circle), DONgav (*n* = 3–4, full grey circle) and DON (*n* = 3–4, full black triangle). The relative gene expression was determined using the cycle threshold (Ct) method and normalized to the expression of the two housekeeping genes (Ct of gene of interest over mean Ct). Differences in unstimulated il-LP explant gene expression level between DONgav vs. DON are displayed with * *p* < 0.05.

**Figure 9 nutrients-12-02115-f009:**
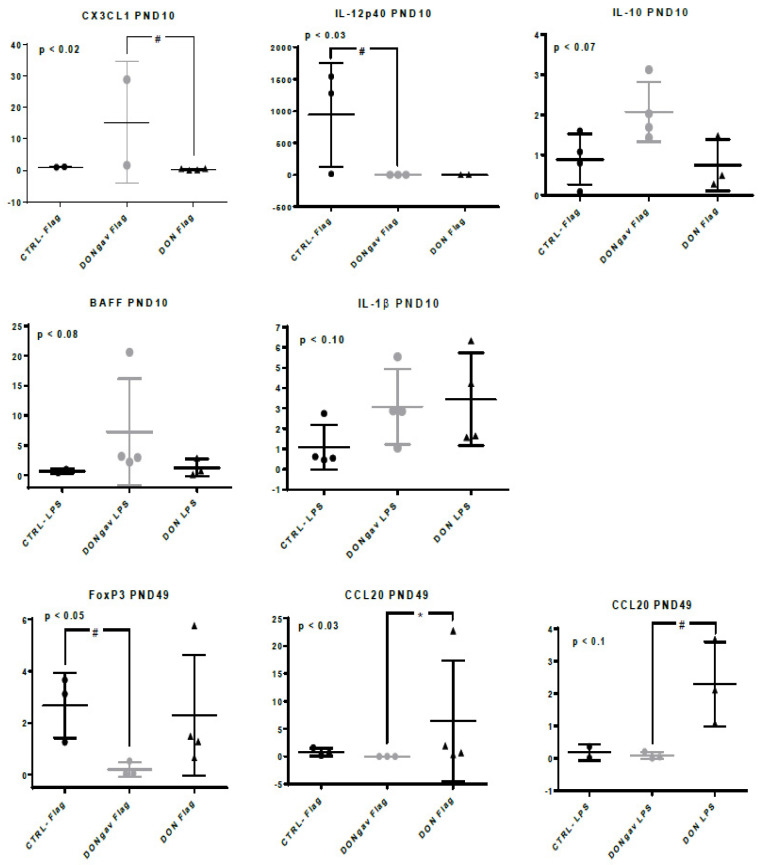
Sow diet mycotoxin contamination (DON) and neonate mycotoxin gavage (DONgav) effect on gene expression of jej-LP explants stimulated with TLR-ligands. Datasets were analyzed using the non-parametric Kruskal-Wallis test (p-value on left corner of each graph) followed by Dunn’s multiple comparison tests when appropriate. Jej-LP explants of CTRL- (*n* = 2–4, full black circle), DONgav (*n* = 2–4, empty grey circle) and DON (*n* = 2–4, full black triangle) were stimulated with flagellin (1 µg/mL) or with LPS (50 µg/mL) at PND10, PND21 and PND49. Relative expressions of the target genes were determined using the 2^−ΔΔCt^ method. Differences in stimulated jej-LP explant gene expression level between CTRL-, DONgav vs. DON are displayed with * *p* < 0.05 or # *p* < 0.1.

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
