# Peer review of "Maternal Supplementation of Food Ingredient (Prebiotic) or Food Contaminant (Mycotoxin) Influences Mucosal Immune System in Piglets"

_nutrients, 2020, doi:10.3390/nu12072115_

Round 1
Reviewer 1 Report
The impact of antenatal nutritional interventions on the development of the immune system is a fascinating topic that is of much interest to researchers currently. Prebiotic fibres are an enticing biologic substrate, and are becoming increasingly widely studied in relation to alter their immunologic effects.
The presented work focuses on antenatal supplementation of a fibre, short-chain fructooligosaccharide (FOS), and a toxin, mycotoxin deoxynivalenol (DON), and their subsequent effect on offspring in a swine model. This is an ambitious project, and I commend the authors for the major effort that certainly was undertaken to perform this project in a large animal model. However, there are substantial concerns regarding the scientific merits of this project, most concerningly relating to the experimental design.
Major concerns
1) The experimental design as presented in figure 1 and described in the methods does not seem to present adequate control groups for comparisons. Specifically two issues standout, namely comparing non-gavaged animals to gavaged animals, and the comparison of the DON and FOS groups.
Ultimately, this project attempts to compare a number of different relationships. Below are, in my view, these relationships, and individual issues with their appropriateness:
a) Prior to birth, the effect of FOS supplementation and DON exposure on sows compared to a control diet.
There is no explanation for why FOS and DON should be directly compared. These are two interventions, intending to produce disparate results, through presumably different mechanisms. Both may be appropriately compared to the a control diet, but not to each other.
b) Comparing DON-gavaged piglets to sham-gavaged piglets (CTRL-).
These comparisons are appropriate with an adequate control group.
c) Comparing perinatal FOS-supplementation to sham-gavaged piglets (CTRL-).
Gavage produces significant stress to animals. It would therefore be difficult to note whether difference between these two groups are due to FOS, or to the stress of gavage. In short, the two groups were not treated similarly outside of the FOS supplementation, and therefore inferences regarding the impact of the FOS cannot be made.
d) Comparing perinatal DON-exposure to sham-gavaged piglets (CTRL-).
Gavage produces significant stress to animals. It would therefore be difficult to note whether difference between these two groups are due to DON, or to the stress of gavage. In short, the two groups were not treated similarly outside of the DONsupplementation, and therefore inferences regarding the impact of the DON cannot be made.
e) Comparing perinatal DON-exposure to FOS-supplementation.
This comparison places two interventions, both of which have unknown effect on the measured variables, in competition with one another. Any differences therefore, cannot be ascribed to either treatment.
I feel that the utility of this work would be greatly enhanced by the addition of a true control group for the PREB and DON groups, one that does not gavage it's piglets.
2) Caloric intake is often a confounding factor in the studies of this type, and must be addressed. Was any attempt made to assess the caloric intake of the sows? Given the differences in sow performance noted in figure 1, have the authors considered that overall caloric intake may be responsible for some of the subsequent findings, rather than direct effects of the fibre.
3) The statistical analysis of this paper may be inappropriate. It is stated that data were assessed for normality, however, most groups identified would have 4 or fewer samples. Demonstrating deviations from normality with this limited a sample size is very difficult. Therefore, the majority of the analysis should be done using non-parametric tests. Additionally, the use of 'trends' or representing p<0.1 as near significant is worrisome, especially given the massive number of comparison performed in this work.
Minor concerns
1) Non-significant relationships should not have '#' or '*' above them in figures, even when those are defined in the figure legend. At a glance these are very misleading. Further, the definitions of these symbols should be consistent between figures (sometimes # means <0.1, other times <0.01)
2) SEM is the measure of dispersion presented. This is rarely the appropriate choice, and SD may be more appropriate.
3) The figures all present means with SEM bars. Given the low number of data points (4 or less for each comparison) it may be helpful to present individual data points, along side the mean and measure of dispersion for the reader.
4) There are numerous small grammatical errors in the work.
Author Response
Dear Editor,
On the behalf of my co-authors, I am glad to re-submit to Nutrients the revised version of our scientific manuscript entitled: “Maternal supplementation of food ingredient (prebiotic) or food contaminant (mycotoxin) influences mucosal immune system in piglets.”
We thank you and the reviewers for your careful reading of our manuscript. We took the constructive suggestions into account and modified our text, accordingly, in order to clarify the points raised by the reviewers. We made some additional corrections (grammatical and typographical) for errors detected when carefully re-reading our manuscript.
In our “Response to Reviewers” file, we provided a point-by-point response to the reviewers’ critiques. We hope that this new version of our manuscript will meet the criteria for publication in Nutrients as we do think that this original study could be of interest for the readers of Nutrients.
We are looking forward to reading your opinion on this revised version our manuscript.
Sincerely Yours,
Stéphanie Ferret-Bernard

Reviewer 2 Report
The authors have studied a very relevant and interesting phenomenon, which needs more attention. Specifically, the generation of new understanding on the impact of environmental triggers, dietary provided pollutions (like DON) on early life immune development is of great importance to the readers of this journal. Within their study they nicely demonstrated that maternal prebiotic supplementation and mycotoxin contamination can modulate mucosal immune system responsiveness of offspring through different pathways. However, the numbers of animals are relatively small, and the in depth functional characterization is missing.
Can the authors add / comment on the functionality of the APC changes depicted in the model related to DON or prebiotic interventions? In addition, if prebiotics are provided, than one also expects changes within the microbiome. i.e. would it be a possibility to add data regarding SCFA analysis, as well as microbiome changes?
Moreover, knowing that DON affects the intestinal barrier, can the authors elaborate / show effects further in the effects in regard? And how the individual small changes link to the changes in APCs could be further elaborated upon.
The current study and set-up has been nicely displayed, and adds to current understanding of the detrimental effect DON can have on the developing immune system, and is therefore of importance to further understand.
Author Response
Dear Editor,
On the behalf of my co-authors, I am glad to re-submit to Nutrients the revised version of our scientific manuscript entitled: “Maternal supplementation of food ingredient (prebiotic) or food contaminant (mycotoxin) influences mucosal immune system in piglets.”
We thank you and the reviewers for your careful reading of our manuscript. We took the constructive suggestions into account and modified our text, accordingly, in order to clarify the points raised by the reviewers. We made some additional corrections (grammatical and typographical) for errors detected when carefully re-reading our manuscript.
In our “Response to Reviewers” file, we provided a point-by-point response to the reviewers’ critiques. We hope that this new version of our manuscript will meet the criteria for publication in Nutrients as we do think that this original study could be of interest for the readers of Nutrients.
We are looking forward to reading your opinion on this revised version our manuscript.
Sincerely Yours,
Stéphanie Ferret-Bernard
Reviewer 2.
The authors have studied a very relevant and interesting phenomenon, which needs more attention. Specifically, the generation of new understanding on the impact of environmental triggers, dietary provided pollutions (like DON) on early life immune development is of great importance to the readers of this journal. Within their study they nicely demonstrated that maternal prebiotic supplementation and mycotoxin contamination can modulate mucosal immune system responsiveness of offspring through different pathways. However, the numbers of animals are relatively small, and the in depth functional characterization is missing.
Response:
Our study aims to provide the proof of concept that the perinatal environment can modulate the developmental profile of the intestinal immune system and specifically, that the mucosal immune system responsiveness of offspring may occur through different pathways depending of the initiating factors. Therefore, we made the choice to give priority to dynamically address issues and provide data from birth to the post-weaning period instead of focusing at only one time point. The major drawback of our strategy was the relatively small numbers of animals at each time point. Looking back at our results, we think that we were right as we have demonstrated that the mucosal immune responses evolved in a time-dependent manner.
Can the authors add / comment on the functionality of the APC changes depicted in the model related to DON or prebiotic interventions?
Response:
In order to add comment of the functionality of the APC related to DON model, this information has been added to the discussion section:
“Previous paper showed that DON increases the number of CD16+ cells migrating from the lamina propria into the epithelium of the jejunum (Nossol C. et al. 2013). This result can explain the decreased of the resident CD16+MHCII+ APC observed in our work. In the intestine, resident dendritic cells play a crucial role in tolerogenic responses including regulatory T cell induction (Scott C.L. et al. 2011). Exposure to DON also induced inhibition of the co-stimulatory factor CD86 in resident dendritic cells. This can be attributed to regulatory T cells which are able to inhibit CD86 expression in dendritic cells which attenuates dendritic cell maturation and stimulatory function (Mavin E. et al. 2017). In link with this result, recent work of Vignal et al. shown in mice model, DON-induced dysbiosis with an enterobacterial bloom with could contribute to its effects on intestinal inflammation (Vignal C. et al. 2018). This study revealed the decrease of Bacteroidetes and the increase of Proteobacteria in DON exposed mice compared to control mice”.
In addition, if prebiotics are provided, than one also expects changes within the microbiome. i.e. would it be a possibility to add data regarding SCFA analysis, as well as microbiome changes?
Response:
Results concerning microbiota composition and fermentative activity are beyond the topic of the present study. In previous published articles (Le Bourgot C. et al. 2014, Le Bourgot C. et al. 2019) have reported the bacterial fermentative activity by analyzing SCFA concentration in intestinal segments (ileum, caecum and colon) and feces of CTRL- and PREB suckling and weaned piglets. When the microbiota of PREB piglets was exposed to a standard fiber-rich dietary environment after weaning, it was able to produce significantly more SCFA than that of CTRL- piglets. In addition, changes in microbiota composition were observed in PREB suckling piglets. Such a different composition of the microbiota early in life would have influenced immune cell polarization by APC, in particular dendritic cells. The nature of T cell polarizing signals is determined largely by the type of microbial products. We noted these results in the introduction (lines 61-63): “Moreover, modification in the microbiota composition and fermentative activity was observed in the offspring when the sow diet was supplemented with prebiotics (Le Bourgot C. et al. 2014, Le Bourgot C. et al. 2019).”
Moreover, knowing that DON affects the intestinal barrier; can the authors elaborate / show effects further in the effects in regard? And how the individual small changes link to the changes in APCs could be further elaborated upon?
Response:
In order to elaborate hypothesis on the consequence of DON effects on intestine barrier, we add this information in discussion section:
“DON influences the epithelial cell turnover in small intestine (Vignal C. et al. 2018). Moreover, the changes in the laminin production and in epithelium composition by DON can provokes an increase of the pore number in the jejunum, which is the basis for an increased migration of CD16+ cells into the epithelium. Diesing et al. (Diesing A.K. et al 2011) showed that high DON concentrations on the enterocyte border induced cell death and loss of the epithelial barrier integrity”.
The current study and set-up has been nicely displayed, and adds to current understanding of the detrimental effect DON can have on the developing immune system, and is therefore of importance to further understand.
Response:
We would like to sincerely thank the reviewer for this comment. Understanding how the immune system development can be modulated by perinatal environment is of great importance for both short-term and long-term health of animals and humans.
Round 2
Reviewer 1 Report
The authors have updated their manuscript, and it's current updated form is an improvement. The presentation of data is less potentially misleading, and more clearly understood by the reader.
Of my major concerns at the initial review, the majority have been addressed. These include the direct comparison between FOS and DOM, the statistical analysis.
Unfortunately the the major concern of different conditions (specifically gavage) between the control and the experimental groups, could not be remedied. These concerns have been addressed in the discussion, but of course still impact the validity of the science. Ethical and practical considerations are cited as the reasons for the inappropriate controls. Perhaps in the future a simplified study design could be employed, rather than a design that sacrifices the integrity of the scientific conclusions being drawn. Nonetheless, I feel addressing this concern clearly in the discussion does serve to alert the reading sufficiently, and this concern should not disqualify publication.
Minor concerns were for the most part corrected adequately.
Reviewer 2 Report
Dear Authors,
Thank you for addressing all comments